# Identifying children who develop severe chronic kidney disease using primary care records

Lucy Plumb[1,2]*, Manish D. Sinha[3], Timothy Jones[1,4], M. Theresa Redaniel[1,4], Matthew J. Ridd[1], Amanda Owen-Smith[1], Fergus J. Caskey[1], Yoav Ben-Shlomo[1,4]

1 Population Health Sciences, University of Bristol Medical School, Bristol, United Kingdom, 2 UK Renal Registry, UK Kidney Association, Bristol, United Kingdom, 3 Department of Paediatric Nephrology, Evelina London Children's Hospital, London, United Kingdom, 4 NIHR Applied Research Collaboration West (ARC West), University Hospitals Bristol and Weston NHS Foundation Trust, Bristol, United Kingdom

* lucy.plumb@bristol.ac.uk

**Data Availability Statement:** This study is based on data from the Clinical Practice Research Datalink (CPRD) obtained under licence from the UK Medicines and Healthcare products Regulatory

## Abstract

### Background

Understanding whether symptoms suggestive of chronic kidney disease (CKD) are reported to primary care before diagnosis may provide opportunities for earlier detection, thus supporting strategies to prevent progression and improve long-term outcomes. Our aim was to determine whether symptoms/signs or consultation frequency recorded in primary care could predict a subsequent diagnosis of chronic kidney disease in children.

### Methods

We undertook a case-control study within Clinical Practice Research Datalink. Cases were children <21 years with an incident code for severe CKD during the study period (January 2000-September 2018). Controls were matched on age (+/-3 years), sex, and practice-level kidney function testing rate. Conditional logistic regression modelling was used to identify symptoms predictive of severe CKD and differences in consultation frequency in 24- and 6-month timeframes before the index date.

### Results

Symptoms predictive of severe CKD in the 24 months before the index date included growth concerns (OR 7.4, 95% CI 3.5, 15.4), oedema (OR 5.7, 95% CI 2.9, 11.2) and urinary tract infection (OR 3.3, 95% CI 2.1, 5.4); within 6 months of the index date, effect estimates and specificity strengthened although sensitivity decreased. Overall, positive predictive value of symptoms was low. Cases consulted more frequently than controls in both timeframes. In combination, symptoms and consultation frequency demonstrated modest discrimination for CKD (c-statistic after bootstrapping 0.70, 95% CI 0.66, 0.73).

Agency. CPRD contains sensitive confidential anonymised longitudinal medical records of patients registered with contributing primary care practices across the UK. Data for this study were derived from the GOLD dataset and are subject to protocol approval and license agreements. Details of terms and conditions of access to CPRD data can be found at https://www.cprd.com. CPRD linked data were provided under a license that does not permit sharing. All protocols requesting access to CPRD data must be submitted via the Electronic Research Applications Portal (eRAP) at https://www.erap.cprd.com. The authors have had access to the data during the study in accordance with the relevant license agreements, which would be available to other researchers once protocol approval is granted. The interpretation and conclusions contained in this study are those of the authors alone.

**Funding:** L Plumb (Doctoral Research Fellowship 2016-09-055) is funded by the National Institute for Health and Care Research (NIHR) for this project. The funding information should now align with this updated statement.

**Competing interests:** The authors have declared that no competing interests exist.

## Conclusion

Despite increased consultation frequency and several symptoms being associated with severe chronic kidney disease, the positive predictive value of symptoms is low given disease rarity making earlier diagnosis challenging.

## Introduction

Chronic kidney disease (CKD) is a substantial health problem for children and young people (CYP) which is associated with complications including impaired growth [1], bone disease [2], anaemia, and cardiovascular morbidity [3] as well as poorer quality of life compared with healthy children [4]. Progression to long-term kidney failure confers significant morbidity and a higher risk of mortality compared with the healthy pediatric population [5]. Internationally, CKD is a major public health problem, affecting as many as 8 to 18% of the global population [6]. If identified early, treatments may help to slow disease progression [7] and preventative strategies may mitigate the burden of cardiovascular disease seen in childhood and later life [8]. Timely detection also supports planning for kidney transplantation which is considered 'best-practice' for eligible children with kidney failure [9], is associated with superior growth [10], cardiovascular health [11–13], quality of life [14] and survival compared to dialysis [15], and is of cost-benefit [16].

Early detection of CKD has been highlighted as a key objective for kidney care, offering improved access to treatments aimed at preventing or delaying progression and better long-term outcomes [17–19]. In the UK and globally, high proportions of children first present to specialist care with advanced CKD [20, 21]. As kidney disease advances, complications relating to a decline in kidney function and an inability to maintain homeostasis are expected [22]. These complications may cause symptoms which in turn, prompt medical review. Primary care is usually the first point of contact for children whose parents or carers have a health concern; due to its rare occurrence however, clinicians are unlikely to accrue experience in identification and diagnosis. No guidance currently exists on symptoms which could potentially alert primary care professionals to a diagnosis of childhood CKD.

To date, one case-control study has investigated whether clinical signs and symptoms can identify children at risk of CKD [23]. The study, which was based on prevalent children with CKD managed in pediatric centres found that signs and symptoms of kidney disease, abnormal antenatal scans, prior neonatal hospitalisation, and a history of hypertension demonstrated good discrimination for CKD. Due to the prevalent nature of the cohort and specialist setting, it is not clear whether symptoms were present before diagnosis or if they could be used to identify incident children in primary care.

No study has explored whether children with CKD can be identified from primary care attendances. Chronic kidney disease is often considered 'silent' until presentation but no study has tested this hypothesis in children. We hypothesised that symptoms reported to and coded by primary care professionals may act as pragmatic markers to support consideration of and testing for CKD. The aim of this study therefore was to determine whether CYP who are subsequently diagnosed with advanced CKD could be distinguished from the general population based on symptoms and consultation frequency, and whether findings could support the development of a clinical prediction tool.

## Materials and methods

### Study design

A population-based, nested case-control study was performed using Clinical Practice Research Datalink (CPRD) GOLD (protocol 17_202), which uses data from practices using Vision® software. CPRD is a prospective longitudinal database that collects patient-level data from participating UK general practices [24]. Children under 18 years account for approximately 20% of the total cohort [25]. CPRD GOLD comprises of a cohort that is broadly similar to the UK general population in term of age, sex, and ethnicity composition [26, 27]. Ethical approval for use of CPRD data was approved by the Medicines and Healthcare Products Regulatory Agency Independent Scientific Advisory Committee in November 2017.

### Study population

The study population consisted of CYP aged 0–21 years (inclusive) registered with a primary care practice that contributed data between January 1, 2000, and August 31, 2018, who had at least 30 days current registration prior to the index date, to ensure incident cases were captured. Cases were CYP with an initial medical code for severe stages of CKD recorded during the study period. It was hypothesised that at this level of kidney function, children could be symptomatic from a decline in kidney function to the extent of requiring medical review. Codes were included if they described CKD stages four or five (estimated glomerular filtration rate $<30$ml/min/1.73m$^2$); established kidney failure; use, or complications of dialysis; kidney transplantation (S1 Table). The index date was the date of the first case-defining code in the medical record. Up to 20 controls who have never had a severe CKD code recorded in their medical record were selected for each case, matching on sex, age (+/-3 years), and deciles of practice kidney function testing rate, a proxy measure for access to investigations for kidney disease and referral at practice level. Each control inherited the index date of the case. Data for this study were extracted from the CPRD GOLD dataset on October 9, 2018, and accessed for research purposes until November 2021. The research team did not have access to information which could identify individual participants during or after data collection.

### Exposure definitions

The presence of symptoms suggestive of CKD (yes/no) and consultation frequency in the 24 months prior to the index date were examined for cases and controls. A systematic approach to identifying and defining symptom variables was undertaken [28] (S1 File). This timeframe was chosen as it was hypothesised that CKD may have a relatively long latency period during which symptoms develop which could represent opportunities for earlier detection. Examination of clinical features during a shorter timeframe (6 months) was also performed to see whether this changed reported symptoms and effect estimates; we speculated that if a symptom variable were causally associated with CKD, any associations observed would strengthen as time to index date decreased.

In line with other work, a consultation was defined as any face-to-face medical consultation which occurred at the practice [29]. Consultations for immunisations were excluded. Sociodemographic reporting was determined using 2015 English Indices of Multiple Deprivation (IMD) scores, which were based on the practice postcode for patients whose practices consented to linkage. Derived from Census data, the IMD score is an ecological composite measure of relative deprivation based on multiple measures (e.g., housing, employment, crime) [30].

## Analysis

Associations between symptoms and consultation frequency with severe CKD were examined using conditional logistic regression to estimate odds ratios (ORs), 95% confidence intervals (CI) and p-values. Use of a matched design does not completely control for confounding [31] and therefore matching variables were included, with age as a categorical variable ($\leq$10; 11–17; 18–21 years). As sex and testing decile were exactly matched, these variables were omitted from the model. To assess the clinical utility of symptoms and consultation frequency, sensitivity, specificity, and positive likelihood ratios (LRs, sensitivity/1-specificity) were calculated [32, 33]. The positive predictive value (PPV) was estimated using Bayes' theorem [34]; prevalence estimates of severe stages of CKD in children aged <18 years were obtained from the UK Renal Registry [35] and applied to the study population.

**Sensitivity analyses.** Analyses were restricted to cases with no evidence of kidney disease (including codes describing lesser stages of CKD) prior to their index date; these individuals were considered a proxy for late presenting individuals. Analyses were also repeated for individuals under the age of 18 years at index date to understand whether findings differed for a population considered to be children under UK law [36]. Where cases were excluded from an analysis, corresponding matching controls were also excluded.

**Prediction model development.** An *a priori* analysis was undertaken to determine whether clinical features could be used to derive a prediction score. Predictive variables (*p* value<0.05) in logistic regression models adjusted for age, sex, and practice testing decile within 6 months of the index date were included in a multivariable model using backward stepwise selection, with variables removed if the corresponding *p* value (Wald test) was >0.05. Interactions between independent variables with age and sex were tested and their inclusion determined using the likelihood ratio test. Calibration was assessed graphically using a plot of the observed versus predicted probabilities, the calibration slope, and calibration-in-the-large, the average predicted risk compared with the overall event rate (ideal value 1) [37, 38]. Discrimination was assessed using the concordance (C-)statistic (equivalent to area under a receiver operating characteristic curve). To avoid bias caused by model overfitting, the final model was internally validated using bootstrap resampling (1000 samples), with β-coefficients and the C-statistic corrected for the calculated optimism.

## Results

During the study period, there were 340 CYP identified with evidence of an incident severe CKD code in the primary care record, which were matched to 6,800 controls. For each timeframe studied, patients who did not contribute data for the full period under study, either because their current registration period commenced or because a transfer out of their registered practice occurred, were excluded from the analysis. Where cases were excluded, the corresponding matched controls were also removed. In total, 261 cases and 4,392 controls were included in the 24-month timeframe analysis, while 290 cases and 5,126 controls were included in the 6-month timeframe analysis. Baseline characteristics are shown in Table 1. Unsurprisingly the demographics of cases and controls were very similar due to matching. Cases were more likely to be represented in the most two deprived IMD quintiles.

Fig 1A and 1B highlights exclusions to the study cohort. Patients who did not contribute data for the respective study periods were excluded. Where cases were excluded, corresponding matched controls were also removed. As age was derived in whole years, children under one on the index date were excluded from the 6-month analysis.

**Table 1. Demographics of study cohort included in each analysis.**

| Demographics | 6-month analysis | | 24-month analysis | |
|---|---|---|---|---|
| | Cases | Controls | Cases | Controls |
| **N** | 290 | 5,126 | 261 | 4,392 |
| Median age in years at index date (IQR) | 15 (8, 18) years | 14 (9, 17) years | 15 (9, 18) years | 14 (9, 18) years |
| **Age group at index date (%)** | | | | |
| 0–10 years | 90 (31.0) | 1,669 (32.6) | 76 (29.1) | 1,297 (29.5) |
| 11–17 years | 104 (35.9) | 2,187 (42.7) | 98 (37.6) | 1,973 (44.9) |
| 18–21 years | 96 (33.1) | 1,270 (24.8) | 87 (33.3) | 1,122 (25.6) |
| **Sex** | | | | |
| Female | 123 (42.4) | 2,183 (42.6) | 106 (40.6) | 1,816 (41.4) |
| Male | 167 (57.6) | 2,943 (57.4) | 155 (59.4) | 2,576 (58.7) |
| **Country of registered practice (%)** | | | | |
| England | 202 (69.7) | 3,655 (71.3) | 183 (70.1) | 3,134 (71.4) |
| Northern Ireland | 9 (3.1) | 219 (4.3) | 9 (3.5) | 194 (4.4) |
| Scotland | 39 (13.5) | 589 (11.5) | 35 (13.4) | 496 (11.3) |
| Wales | 40 (13.8) | 663 (12.9) | 34 (13.0) | 568 (12.9) |
| **Indices of multiple deprivation quintile** *(based on practice postcode; for practices which consent to linkage)* **(%)** | | | | |
| N included in analysis | | | | |
| 1 (Least Deprived) | 19 (12.6) | 448 (15.9) | 18 (13.1) | 393 (16.3) |
| 2 | 28 (18.5) | 517 (18.3) | 25 (18.3) | 430 (17.8) |
| 3 | 21 (13.9) | 581 (20.6) | 19 (13.9) | 489 (20.3) |
| 4 | 45 (29.8) | 565 (20.0) | 43 (31.4) | 496 (20.6) |
| 5 (Most Deprived) | 38 (25.2) | 711 (25.2) | 32 (23.4) | 606 (25.1) |

Abbreviations: IQR, Interquartile range.

## Prediction modelling using 24-month timeframe

Using age-adjusted conditional logistic regression, eight individual symptom variables were predictive of severe CKD in the 24 months before the index date: change in urine colour, cough, flank pain, oedema, urinary tract infection (UTI), vomiting and growth concerns (including weight loss), as well as generic 'unwell' codes (Table 2). The most sensitive symptoms were cough (23.4%), vomiting (11.9%) and UTI (9.2%). Cough was the least specific

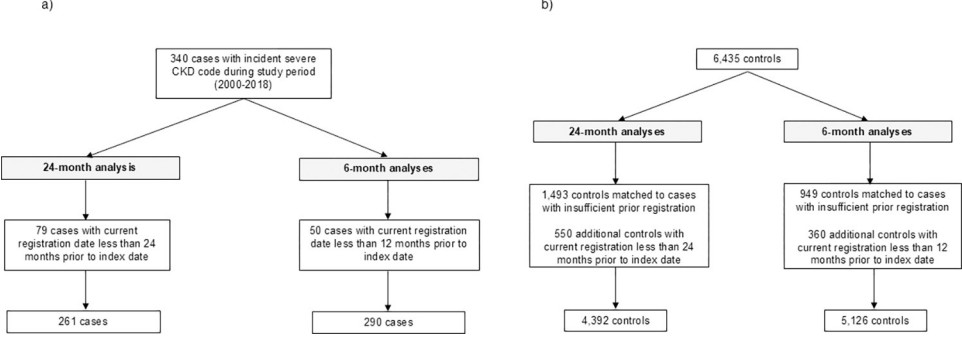

**Fig 1.** Exclusions to the study cohort for cases (a) and controls (b). Footnote: Where cases were excluded, corresponding matched controls were also removed. As age was derived in whole years, children under one year on the index date were excluded from the 6-month analysis.

**Table 2. Frequency of symptoms predictive of severe CKD between cases and controls in the 24 and 6 months prior to the index date.**

| | Controls | | Cases | Diagnostic utilities in the 24 months prior to the index date | | | | Conditional regression |
|---|---|---|---|---|---|---|---|---|
| | N | % | N | Sensitivity | Specificity | PPV per 1000 (95% CI) | LR (95% CI) | OR (95% CI) |
| **24 months prior to index date** | | | | | | | | |
| | 4,392 | | 261 | | | | | 4,653 |
| Cough | 763 | 17 | 61 | 23 | 83 | 0.1 (0.1, 0.2) | 1.4 (1.1, 1.7) | 1.46 (1.07, 1.98) |
| Flank pain | -* | -* | -* | -* | 100 | 1.7 (0.1, 26.1) | 16.8 (1.1, 268.0) | 17.2 (1.05, 280) |
| Generally unwell | 44 | 1 | 8 | 3 | 99 | 0.3 (0.1, 0.6) | 3.1 (1.5, 6.4) | 3.17 (1.48, 6.79) |
| Growth | 26 | 0.6 | 11 | 4 | 99 | 0.7 (0.4, 1.4) | 7.1 (3.6, 14.2) | 7.35 (3.52, 15.4) |
| Oedema | 34 | 0.8 | 12 | 5 | 99 | 0.6 (0.3, 1.1) | 5.9 (3.1, 11.3) | 5.72 (2.92, 11.2) |
| Urine colour | 10 | 0.2 | -* | -* | 100 | 0.7 (0.2, 2.1) | 6.7 (2.1, 21.3) | 6.52 (2.03, 21.0) |
| UTI | 136 | 3 | 24 | 9 | 97 | 0.3 (0.2, 0.4) | 3.0 (2.0, 4.5) | 3.32 (2.05, 5.38) |
| Vomiting | 192 | 4 | 31 | 12 | 96 | 0.3 (0.2, 0.4) | 2.7 (1.9, 3.9) | 2.95 (1.96, 4.43) |
| **6 months prior to index date** | | | | | | | | |
| | 5,126 | | 290 | | | | | 5,416 |
| Generally unwell | 12 | 0.2 | -* | -* | 100 | 0.6 (0.2, 1.8) | 5.9 (1.9, 18.2) | 6.25 (1.97, 19.8) |
| Growth | 6 | 0.1 | 6 | 2 | 100 | 1.8 (0.6, 5.4) | 17.7 (5.7, 54.5) | 17.1 (5.46, 53.5) |
| Headache | 86 | 2 | 11 | 4 | 98 | 0.2 (0.1, 0.4) | 2.3 (1.2, 4.2) | 2.32 (1.21, 4.46) |
| Oedema | 9 | 0.2 | 5 | 2 | 100 | 1.0 (0.3, 2.9) | 9.8 (3.3, 29.1) | 8.36 (2.74, 25.5) |
| SOB | 44 | 0.9 | 6 | 2 | 99 | 0.2 (0.1, 0.6) | 2.4 (1.0, 5.6) | 2.44 (1.02, 5.83) |
| Thirst | 5 | 0.1 | -* | -* | 100 | 0.7 (0.1, 3.6) | 7.1 (1.4, 36.3) | 6.41 (1.24, 33.2) |
| UTI | 42 | 0.8 | 9 | 3 | 99 | 0.4 (0.2, 0.8) | 3.8 (1.9, 7.7) | 3.66 (1.74, 7.71) |
| Urine colour | -* | -* | -* | -* | 100 | 1.8 (0.2, 1.2) | 17.7 (2.5, 125.0) | 18.1 (2.51, 130.8) |
| Visual problems | 5 | 0.1 | -* | -* | 100 | 0.7 (0.1, 3.6) | 7.1 (1.4, 36.3) | 6.50 (1.24, 34.05) |
| Vomiting | 69 | 1 | 16 | 6 | 99 | 0.4 (0.2, 0.7) | 4.1 (2.4, 7.0) | 4.14 (2.36, 7.26) |

Abbreviations: CI, confidence interval; LR, positive likelihood ratio; PPV, positive predictive value. *Cells containing fewer than 5 cases are suppressed in line with Clinical Practice Research Datalink's policy. Note: PPV per 1000 implies of 1000 children coded, 2 would be diagnosed with severe chronic kidney disease.

(82.6%), followed by vomiting (95.6%). In this timeframe, the highest positive likelihood ratios and predictive values were seen for flank pain (LR 16.8, 95% CI 1.1, 268.0; PPV 1.7 per 1000) and growth concerns (LR 7.1, 95% CI 3.6, 14.2; PPV 0.7 per 1000).

## Prediction modelling using 6-month timeframe

Within 6 months of the index date, codes describing a generally unwell child, growth failure, oedema, change in urine colour, UTI and vomiting remained positively associated with the outcome, while headache, dyspnoea/shortness of breath, thirst and visual problems were individual symptoms also predictive of severe CKD (Table 2). The most sensitive codes were vomiting (5.5%), headache (3.8%) and UTI (3.1%). All symptoms demonstrated high specificity (>98%). Growth concerns and change in urine colour had the highest LRs (17.7, 95% CI 5.7, 54.5 and 17.7, 95% CI 2.5, 125.0, respectively) and PPVs (1.8 per 1000 children e.g., of 1,000 children coded, 2 would be diagnosed with severe CKD in 6 months), followed by oedema (LR 9.8, 95% CI 3.3, 29.1; PPV 1.0 per 1000). Colour change was more frequently reported by cases but was not predictive of severe CKD in regression models; codes for oliguria and flank pain were seen among cases only.

To further investigate symptom reporting, associations with severe CKD were examined in the 0–6 and >6–24 months prior to diagnosis (S1 Fig). In the >6–24 months prior to diagnosis, growth concerns, vomiting, UTI, and oedema remained predictive of case status.

## Sensitivity analyses

When children with evidence of known kidney disease were excluded, oedema, growth, vomiting, and vision independently remained strongly associated with case status in both 24- and 6-month timeframes (S2 Table). Similar findings were observed when the analysis was restricted to under 18-year-olds, although effect estimates were attenuated (S3 Table).

## Consultation frequency

Higher odds of being a case were noted with increasing consultation frequency (Table 3). A similar pattern was seen when children with known kidney disease or ≥18-year-olds were excluded.

## Risk score development

The final multivariable model included codes for growth concerns, being 'generally unwell', oedema, vomiting, UTI, urine colour and consultation frequency as an ordinal variable. As the symptom 'generally unwell' was considered vague and difficult to define, it was excluded from the final model. There was evidence of effect modification (interaction) between age and growth reporting, with higher odds of severe CKD seen in younger children reporting growth concerns compared to older children. Sex was noted to modify the effect of oedema reporting on severe CKD, with higher odds of severe CKD among females with oedema present. Interaction terms for both were therefore included in the final model. Table 4 highlights findings from the prediction model, before and after internal validation, with mutual adjustment for included variables. A plot of the observed versus the expected probabilities demonstrated acceptable calibration, with a calibration slope of 0.91 (95% CI 0.00, 1.11) and calibration-in-the-large of -0.01 (95% CI -0.12, 0.12, see S2 Fig). The β-coefficients of the bootstrapped model

**Table 3. Associations between consultation frequency and severe CKD within the 24 and 6 months before the index date.**

| Consultation frequency | Controls | | Cases | | | Conditional logistic regression analysis (Age-adjusted) | | |
|---|---|---|---|---|---|---|---|---|
| | N | % | N | % | | OR | 95% CI | |
| **24 months prior to index date** | | | | | | | | |
| | 4,392 | | 261 | | | 4,653 | | |
| 0 | 492 | 11 | 16 | 6 | | Reference | | |
| 1 | 476 | 11 | 8 | 3 | | 0.52 | 0.22 | 1.22 |
| 2 | 486 | 11 | 19 | 7 | | 1.26 | 0.64 | 2.48 |
| 3 | 439 | 10 | 15 | 6 | | 1.07 | 0.52 | 2.2 |
| ≥4 | 2,499 | 57 | 203 | 78 | | 2.55 | 1.51 | 4.31 |
| | | | | | As ordinal variable | 1.41 | 1.25 | 1.58 |
| **6 months prior to index date** | | | | | | | | |
| | 5,126 | | 290 | | | 5,416 | | |
| 0 | 2,510 | 49 | 77 | 27 | | Reference | | |
| 1 | 1,063 | 21 | 40 | 14 | | 1.61 | 1.05 | 2.47 |
| 2 | 637 | 12 | 37 | 13 | | 2.63 | 1.69 | 4.09 |
| 3 | 347 | 7 | 32 | 11 | | 4.02 | 2.52 | 6.42 |
| ≥4 | 569 | 11 | 104 | 36 | | 8.33 | 5.78 | 12.02 |
| | | | | | As ordinal variable | 1.7 | 1.56 | 1.85 |

Note: Likelihood ratio testing indicates better goodness of fit with consultation frequency modelled as continuous variable.

**Table 4. Multivariable prediction model for severe CKD based on presenting symptomatology to primary care in previous 6 months (n = 5,416, 290 cases).**

| Variable | Odds Ratio before internal validation (95% CI) | Odds Ratio after internal validation (95% CI) | Adjusted coefficient for prediction score |
|---|---|---|---|
| **Growth concern (interaction with age)** *p* value for interaction 0.01 | | | |
| **0–10 years** | | | |
| Concern present | 36.5 (1.5, 880.3) | 20.5 (1.4, 293.5) | 3.0 |
| Concern absent | 0.8 (0.6, 1.1) | 0.8 (0.6, 1.1) | -0.2 |
| **11–17 years** | | | |
| Concern present | 3.6 (0.1, 130.0) | 2.9 (0.1, 59.1) | 1.1 |
| Concern absent | 0.8 (0.6, 1.0) | 0.8 (0.6, 1.1) | -0.2 |
| **18–21 years** | | | |
| Concern present | 2.6 (0.3, 25.1) | 2.2 (0.3, 14.9) | 0.8 |
| Concern absent | Reference | Reference | 0.0 |
| **Oedema (interaction with sex)** *p* value for interaction 0.03 | | | |
| **Female** | | | |
| Concern present | 26.8 (4.6, 156.0) | 15.7 (3.6, 68.8) | |
| Concern absent | Reference | Reference | 0.0 |
| **Male** | | | |
| Concern present | 0.1 (0.0, 1.3) | 0.1 (0.01, 1.2) | 0.2 |
| Concern absent | 1.2 (0.9, 1.6) | 1.2 (0.9, 1.4) | -2.1 |
| **UTI** | 2.43 (1.1, 5.2) | 2.24 (1.12, 4.48) | 0.7 |
| **Urine colour** | 12.1 (1.5, 97.7) | 9.57 (1.44, 63.50) | 2.1 |
| **Vomiting** | 2.6 (1.4, 4.6) | 2.35 (1.38, 4.01) | 0.8 |
| **Frequency** *Per consultation, up to $\geq 4$* | 1.5 (1.4, 1.7) | 1.47 (1.37, 1.58) | 0.4 |
| **Constant** | 0.03 (0.02, 0.04) | 0.03 (0.03, 0.04) | -3.4 |
| **C-statistic** | 0.70 (0.67, 0.74) | 0.70 (0.66, 0.73) | |

Abbreviations: CI, confidence interval; UTI, urinary tract infection. Model internally validated using bootstrap resampling (1000 samples). Risk score = cons+ $\beta_1$(consfreq) + $\beta_2$(uti)+ $\beta_3$(urinecolour)+ $\beta_4$(vomiting)+ $\beta_5$(sex)+$\beta_6$(sex*oedema) + $\beta_7$(ageband)+ $\beta_8$(ageband*growth). β-coefficients for (ageband)+(ageband*growth) and (sex)+(sex*growth) presented for each strata. Predicted risk = exp(riskscore)/(1+exp(riskscore))

were used assign a risk score and calculate predicted risk (equation shown in legend of Table 4). A risk threshold of 10% or more offered high positive likelihood ratios (S4 Table; LR at 10% predicted risk: 2.8, 95% CI 2.0, 3.7) and high specificity (94.7%, 95% CI 94.1, 95.3) although positive predictive values remained low due to the low background disease prevalence.

## Discussion

To our knowledge, this is the first study to use national primary care data to investigate the utility of symptom reporting and consultation patterns in predicting childhood CKD. Growth concerns, oedema, UTI, and vomiting were more common among cases and predictive of subsequent severe CKD in the 24-month period prior to the index date, challenging the notion of CKD as a 'silent' disease. Individual symptoms had a low PPV due to the low disease incidence. While the symptoms identified were not sensitive, their specificity was generally high and increased closer to the index date. Children who went on to develop severe CKD also consulted more frequently with a primary care clinician in the lead up to diagnosis. In an

exploratory analysis, symptoms along with consultation frequency demonstrated moderate discrimination for severe CKD with a regression model c-statistic of 0.70 (95% CI 0.66, 0.73).

## Comparison with existing literature

In our review of the literature, there is only one published risk score supporting CKD identification in children [23]. This identified similar symptoms, particularly growth concerns, oedema, and UTI, were predictive of disease compared to controls, despite differences in setting and study population. Our results are also consistent with observational studies of children referred to pediatric nephrology care, with several noting prolonged symptomatic periods before presentation [20, 39, 40]. From these findings however, it was not clear whether patients were previously seen by other healthcare providers and where interventions could be targeted. We confirm that symptoms suggestive of CKD are not only present before diagnosis but may be detectable for some time beforehand, highlighting potential opportunities for earlier diagnosis and intervention.

## Strengths and limitations

This study was inclusive of all cases with severe stages of CKD identified in CPRD GOLD, a representative UK primary care database, which ensures results are generalisable to all children with CKD irrespective of underlying disorder as well as being applicable to other high-income countries with primary healthcare services. Symptom code lists were compiled using established methodology as has been employed in other studies [32, 41–43]. Furthermore, exposure status was assigned based on symptoms documented prospectively prior to coding of severe CKD which helps minimise information bias.

There are however limitations. As determination of symptom exposure was reliant on practitioners recording symptoms in the electronic patient record, under-coding may have led to underestimation of effect estimates, assuming that any misclassification is non-differential to future case status. Given similar predictive symptoms were noted when cases with pre-existing kidney codes were excluded, we are assured differential misclassification of symptom exposure did not occur; this finding also makes it less likely that reverse causality (children with advanced CKD are more likely to be coded for CKD-specific symptoms) has occurred. Due to the number of controls per case required, we were unable to match on more granular variables such as practice, raising the possibility of residual confounding.

## Implications for research and/or practice

While there is increasing focus on early CKD recognition to prevent or delay kidney disease progression, it is not clear what the best strategy is to maximise detection whilst minimising harm, costs, and inappropriate use of resources. In adults, evidence suggests that a 'case finding' approach, screening patients known to be at higher risk of CKD due to coexisting conditions, is a more efficient option to population screening [18, 19]. For children, there are sparse data regarding who to test for CKD, despite the potential health and economic gains to be made across their lifecourse. In 2007, the American Academy of Pediatrics recommended discontinuation of routine urinalysis screening in children, a practice which Sekhar and colleagues estimated to be cost-ineffective [44]. Currently UK guidance from the National Institute for Health and Care Excellence advise consideration of blood and urine testing for children at risk of CKD development including those with diabetes, hypertension or known structural kidney disease, although acknowledge the limited evidence base [45]. In their 2021 evidence update, the national body stated research identifying prognostic factors for CKD development in children was a high national priority [46]. This study has demonstrated that

pragmatic markers such as symptoms and consultation frequency can be used to identify children who go on to develop severe CKD. Incorporating a prediction model into clinical systems may support decision-making regarding testing and referral, which has helped shorten time to diagnosis for other diagnostically challenging and rare conditions such as childhood brain tumours [47]. Considering the low PPV for individual symptoms, it would be inappropriate for primary care clinicians to refer all children with these symptoms to specialist care for investigation, however the high specificity of symptoms, both individually and in combination is reassuring. For children with high predicted risk, urinalysis, or blood tests may help to confirm or refute a diagnosis of CKD. While this may increase costs associated with testing, identifying CKD in earlier stages may result in a more cost-effective pathway for future management and monitoring, thus representing appropriate use of healthcare resources [46]. Further work is needed to determine the economic impact on incorporating such a clinical tool into practice. In addition, work is needed to determine whether the discriminatory power of this model can be enhanced with inclusion of other variables readily accessible to primary healthcare teams before external validation [48]. Future mixed-methods research should explore the acceptability of a clinical prediction score as well as appropriate risk thresholds for invasive and non-invasive testing in children, from the perspectives of the child, family, and professionals.

## Conclusions

Key symptoms and consultation frequency documented in the primary care record are associated with subsequent coding for severe CKD in CYP. Individual symptoms have a low positive predictive value due to the rarity of kidney disease among the pediatric population. In combination, symptoms, with consultation frequency, age, and sex, may be used to develop a clinical prediction score though this has modest discrimination for severe CKD. Further work is now needed to quantify the potential economic and health gains of such a model to support earlier diagnosis for children, families, and healthcare providers.

## Supporting information

**S1 Checklist. STROBE statement—Checklist of items that should be included in reports of *case-control studies.***
(DOC)

**S2 Checklist. TRIPOD checklist: Prediction model development.**
(DOCX)

**S1 Table. Read codes used to describe severe chronic kidney disease.**
(PDF)

**S2 Table. The association of predictive symptoms and a subsequent severe CKD code between cases and controls with no prior evidence of kidney disease, 24 and 6 months prior to index date.**
(PDF)

**S3 Table. The association of predictive symptoms and a subsequent severe CKD code between cases and controls under 18 years of age, 24 and 6 months prior to index date.**
(PDF)

**S4 Table. Test characteristics at ascending cut-off points of clinical risk prediction score for severe CKD.**
(PDF)

**S1 File. Symptom/sign exposure variable codelists.**
(PDF)

**S1 Fig. Symptom associations with severe stages of CKD in the >6–24 and 0-≥6 months prior to the index date, between cases and controls.**
(TIF)

**S2 Fig. Predicted and observed probabilities of having severe CKD within 6-month timeframe.**
(TIF)

## Acknowledgments

We thank Dr Stephanie MacNeill for her advice regarding study design and statistical analysis. Findings from this study were presented in abstract form at UK Kidney Week 2022. The views expressed are those of the authors and not necessarily those of the NHS, the National Institute for Health and Care Research (NIHR) or the UK Department of Health and Social Care.

## Author Contributions

**Conceptualization:** Lucy Plumb, Manish D. Sinha, Timothy Jones, M. Theresa Redaniel, Fergus J. Caskey, Yoav Ben-Shlomo.

**Data curation:** Lucy Plumb, Manish D. Sinha, Timothy Jones, M. Theresa Redaniel, Matthew J. Ridd, Amanda Owen-Smith, Fergus J. Caskey, Yoav Ben-Shlomo.

**Formal analysis:** Lucy Plumb, Manish D. Sinha, Matthew J. Ridd, Fergus J. Caskey, Yoav Ben-Shlomo.

**Funding acquisition:** Lucy Plumb.

**Methodology:** Lucy Plumb, Manish D. Sinha, Timothy Jones, M. Theresa Redaniel, Matthew J. Ridd, Amanda Owen-Smith, Fergus J. Caskey, Yoav Ben-Shlomo.

**Project administration:** Lucy Plumb.

**Supervision:** Manish D. Sinha, Amanda Owen-Smith, Fergus J. Caskey, Yoav Ben-Shlomo.

**Writing – original draft:** Lucy Plumb.

**Writing – review & editing:** Lucy Plumb, Manish D. Sinha, Timothy Jones, M. Theresa Redaniel, Matthew J. Ridd, Amanda Owen-Smith, Fergus J. Caskey, Yoav Ben-Shlomo.

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
