## [Decision Letter · Decision Letter 0]

7 Jun 2024

PONE-D-24-05195Identifying children who develop severe chronic kidney disease using primary care recordsPLOS ONE

Dear Dr. Plumb,

Thank you for submitting your manuscript to PLOS ONE. After careful consideration, we feel that it has merit but does not fully meet PLOS ONE’s publication criteria as it currently stands. Therefore, we invite you to submit a revised version of the manuscript that addresses the points raised during the review process.

We look forward to receiving your revised manuscript.

Kind regards,

Rajendra Bhimma, PhD

Academic Editor

PLOS ONE

Journal Requirements:

Additional Editor Comments:

Dear Dr Lucy Plumb.

Thank you for the submission of your manuscript. It has been reviewed by two separate experts in the field and there are some concerns raised that need to be addressed.

Reviewers' comments:

Reviewer's Responses to Questions

**Comments to the Author**

1. Is the manuscript technically sound, and do the data support the conclusions?

Reviewer #1: Yes

Reviewer #2: Yes

2. Has the statistical analysis been performed appropriately and rigorously? 

Reviewer #1: Yes

Reviewer #2: I Don't Know

3. Have the authors made all data underlying the findings in their manuscript fully available?

Reviewer #1: Yes

Reviewer #2: Yes

4. Is the manuscript presented in an intelligible fashion and written in standard English?

Reviewer #1: Yes

Reviewer #2: Yes

5. Review Comments to the Author

Reviewer #1: Here are my comments,

With exception of Table 2 and row 240, it is a bit unclear if the PPV is calculated for each type of symptoms or for combination of symptoms.

Row 14 - On the conclusion “positive predictive value of symptoms is low”, is the conclusion rather that “In combination, symptoms and consultation frequency demonstrated modest discrimination for CKD”?

Row 135 – On “there were 340 CYP with evidence of an incident severe CKD code in the primary care record, which were matched to 6,800 controls”, it would be more interesting with the numbers of CYP and controls that was used in the 24 months analysis and the 6 months analysis respectively.

Row 139 – Rater present the demographics table for the subjects included in the 24 months analysis and the 6 months analysis respectively.

Row - 163 Table 2, The heading “PPV per 1000” may be hard to understand, perhaps add a footnote of type ‘PPV 1.8 per 1000’ implies of 1,000 children coded, 2 would be diagnosed with severe CKD”.

Figure 1 – For the “6-month analyses” in panel a and b, should the text “registration less than 12 months” rather be “registration less than 6 months”?

Reviewer #2: It is a very relevant topic as it would begin monitoring & reversing some of the factors that can be reversed in chronic renal failure e.g. treatment of urinary tract infection diagnosis and management of hypertension, dietary manupulation, management of short stature and failure to thrive and delaying the onset of renal replacement therapy in theses chronic conditions.

To simplify the scoring system. I recognize that this was a study but for the general practitioners & general paediatricians, we need a simple algorithm.

It is a very relevant study.

6. PLOS authors have the option to publish the peer review history of their article (what does this mean?). If published, this will include your full peer review and any attached files.

Reviewer #1: No

Reviewer #2: No

---

## [Author Response · Author response to Decision Letter 0]

22 Jul 2024

We thank the reviewers for the insightful and constructive comments. We hope to have addressed the comments satisfactorily. 

Review Comments to the Author

Reviewer #1: Here are my comments,

With exception of Table 2 and row 240, it is a bit unclear if the PPV is calculated for each type of symptoms or for combination of symptoms.

Thank you for this comment. We have now made clear in the text where we refer to individual symptoms; we have also highlighted that in the prediction model there is mutual adjustment for the included variables. 

Row 14 - On the conclusion “positive predictive value of symptoms is low”, is the conclusion rather that “In combination, symptoms and consultation frequency demonstrated modest discrimination for CKD”?

Thank you for this comment. We believe both statements to be accurate, as evidenced by the individual symptoms and their associations with case status (table 2), and as evidenced by the C-statistic in table 4 for symptoms and consultation frequency combined. 

Row 135 – On “there were 340 CYP with evidence of an incident severe CKD code in the primary care record, which were matched to 6,800 controls”, it would be more interesting with the numbers of CYP and controls that was used in the 24 months analysis and the 6 months analysis respectively.

We have now amended this section to refer directly to the study cohort used in the two analyses (24 months and 6 months, respectively). 

Row 139 – Rater present the demographics table for the subjects included in the 24 months analysis and the 6 months analysis respectively.

Please find table one has now been amended to reflect the study cohorts used. 

Row - 163 Table 2, The heading “PPV per 1000” may be hard to understand, perhaps add a footnote of type ‘PPV 1.8 per 1000’ implies of 1,000 children coded, 2 would be diagnosed with severe CKD”.

Thank you for this suggestion. We have added this as a footnote for ease of interpretation. 

Figure 1 – For the “6-month analyses” in panel a and b, should the text “registration less than 12 months” rather be “registration less than 6 months”?

As age was derived in whole years, children under one on the index date were excluded from the 6-month analysis. This is also referenced in the text (lines 144-145). 

Reviewer #2: It is a very relevant topic as it would begin monitoring & reversing some of the factors that can be reversed in chronic renal failure e.g. treatment of urinary tract infection diagnosis and management of hypertension, dietary manipulation, management of short stature and failure to thrive and delaying the onset of renal replacement therapy in theses chronic conditions.

To simplify the scoring system. I recognize that this was a study but for the general practitioners & general paediatricians, we need a simple algorithm.

Thank you for this suggestion. We present the risk score as a footnote to table 4. We are planning to externally validate this algorithm before developing a risk calculator for ease of use clinically.

---

## [Decision Letter · Decision Letter 1]

5 Nov 2024

Identifying children who develop severe chronic kidney disease using primary care records

PONE-D-24-05195R1

Dear Dr. Plumb,

We’re pleased to inform you that your manuscript has been judged scientifically suitable for publication and will be formally accepted for publication once it meets all outstanding technical requirements.

Kind regards,

Amjad Khan, Ph.D.

Academic Editor

PLOS ONE

Additional Editor Comments (optional):

Reviewers' comments:

Reviewer's Responses to Questions

**Comments to the Author**

1. If the authors have adequately addressed your comments raised in a previous round of review and you feel that this manuscript is now acceptable for publication, you may indicate that here to bypass the “Comments to the Author” section, enter your conflict of interest statement in the “Confidential to Editor” section, and submit your "Accept" recommendation.

Reviewer #1: All comments have been addressed

Reviewer #2: All comments have been addressed

2. Is the manuscript technically sound, and do the data support the conclusions?

Reviewer #1: Yes

Reviewer #2: Yes

3. Has the statistical analysis been performed appropriately and rigorously? 

Reviewer #1: Yes

Reviewer #2: I Don't Know

4. Have the authors made all data underlying the findings in their manuscript fully available?

Reviewer #1: No

Reviewer #2: Yes

5. Is the manuscript presented in an intelligible fashion and written in standard English?

Reviewer #1: Yes

Reviewer #2: Yes

6. Review Comments to the Author

Reviewer #1: Thank you for your response and revision.

Thank you for your response and revision.

Thank you for your response and revision.

Reviewer #2: It is a very important topic as detection of chronic renal disease earlier results in trying to preserve renal function long as possible, delying institution of renal replacement therapy.

7. PLOS authors have the option to publish the peer review history of their article (what does this mean?). If published, this will include your full peer review and any attached files.

Reviewer #1: No

Reviewer #2: No

---

## [Editor Report · Acceptance letter]

16 Nov 2024

PONE-D-24-05195R1 

PLOS ONE

Dear Dr. Plumb, 

I'm pleased to inform you that your manuscript has been deemed suitable for publication in PLOS ONE. Congratulations! Your manuscript is now being handed over to our production team.

Kind regards, 

on behalf of

Dr. Amjad Khan 

Academic Editor

PLOS ONE